# Properties of Magnetic Garnet Films for Flexible Magneto-Optical Indicators Fabricated by Spin-Coating Method

**DOI:** 10.3390/ma15031241

**Published:** 2022-02-07

**Authors:** Ryosuke Hashimoto, Toshiya Itaya, Hironaga Uchida, Yuya Funaki, Syunsuke Fukuchi

**Affiliations:** 1Department of Electronic and Information Engineering, National Institute of Technology, Suzuka College, Shiroko-cho, Suzuka 510-0294, Mie, Japan; itaya@info.suzuka-ct.ac.jp (T.I.); h29e37@ed.cc.suzuka-ct.ac.jp (S.F.); 2Department of Electrical and Electronic Information Engineering, Toyohashi University of Technology, Hibarigaoka, Tempaku-cho, Toyohashi 441-8580, Aichi, Japan; uchida@ee.tut.ac.jp (H.U.); funaki.yuya.xk@tut.jp (Y.F.)

**Keywords:** magneto-optical imaging, nondestructive testing, magnetic garnet, spin coating method, transmittance spectrum

## Abstract

Non-destructive testing using a magneto-optical effect is a high-resolution non-destructive inspection technique for a metallic structure. It is able to provide high-spatial resolution images of defects. Previously, it has been difficult to fabricate flexible magneto-optical sensors because thermal treatment is necessary to crystallize the magnetic garnet. Therefore, it was not possible to apply magneto-optical imaging to complicated shapes in a test subject, such as a curved surface. In this study, we developed a new process for deposition of the magnetic garnet on the flexible substrate by applying the magnetic garnet powders that have already undergone crystallization. In this new process, as it does not require thermal treatment after deposition, flexible substrates with low heat resistance can be used. In this paper, we report our observations of the optical properties, magnetic hysteresis loop, crystallizability and density of the particles on the flexible substrate deposited by the spin-coating method.

## 1. Introduction

Currently, safety is maintained by detecting defects in a structure at an early stage with various non-destructive testing methods. Non-destructive testing using a magneto-optical (MO) effect is a high-resolution non-destructive inspection technique for a metallic structure [1,2,3,4,5,6]. The MO effect is a phenomenon where a polarization plane of light rotates when a linearly polarized light passes through the magnetized magnetic materials [7]. MO imaging is performed by placing a thin film, wherein a magnetic garnet is layered on a substrate on a test subject as an MO sensor [8,9,10,11].

Previously, the deposition process of the MO sensor required high-temperature thermal treatment [12] to crystallize the magnetic garnet because a sputtering method was used for the deposition of the magnetic garnet film [13,14]. Therefore, it was limited to a single crystalline substrate that was not able to deal with complicated shapes, such as a curved surface, which was contained in the test subject.

The purpose of our research is to enable the magnetic garnet film to form on flexible substrates and evaluate its physical properties in order to manufacture a flexible MO sensor capable of coping with curved surfaces. In our research group, we looked at a spin-coating method as the film formation method of the magnetic garnet. There are several methods to prepare the magnetic garnet films by using spin-coating methods. The metal-organic decomposition method [15,16,17,18,19,20,21,22,23] and the sol-gel method [24,25,26,27,28,29,30] are generally used, but heat treatment at the temperature from 600 to 900 °C. is necessary as with the sputtering method. Therefore, we developed a new process for deposition of the magnetic garnet materials on the flexible substrate by applying the magnetic garnet powders that have already undergone crystallization. In this new process, as it does not require thermal treatment after deposition, flexible substrates with low heat resistance can be used.

In this paper, we report the observation results of the optical properties, magnetic hysteresis loop, crystallizability and density of the particles on the flexible substrate deposited by the spin-coating method.

## 2. Materials and Methods

The MO effect is a phenomenon related to the polarization state of the transmitted or the reflected light of a magnetized material. This effect contains some varieties, and this study describes MO imaging using the Faraday effect, which is the MO effect for light transmitted through a magnetic material. Figure 1 shows a schematic illustration of the Faraday effect. The Faraday effect is a magnetic phenomenon where a polarization plane rotates when linearly polarized light passes through a magnetic material. The angle of the polarization plane rotated due to the Faraday effect is referred to as the Faraday rotation angle (*θ*), and its magnitude is given in Equation (1).
*θ* = *F* (*M*/*MS*) *L*(1)

In Equation (1), *M**S* is the saturation magnetization of a magnetic material, *M* is the magnitude of magnetization parallel to the traveling direction of light, and *L* is the distance where light passes through (i.e., the thickness of the magnetic material). It should be noted that *F* is a value specific to a material called a Faraday rotational coefficient. As seen in Equation (1), the Faraday rotational angle increases in proportion to the thickness of the magnetic material and the magnitude of magnetization.

The light intensity (*I*_1_) of the MO image generated with an orthogonal analyzer method (Figure 1) is expressed as in the following Equation (2), where *I*_0_ is the light intensity of the incident light, %*T* is the transmittance of the MO material, and *θ* is the Faraday rotational angle of the MO material.
(2)$$I_{1}=I_{0}\times \%T\times \sin^{2}\left(\theta \right)$$

Therefore, it is important that the film has high transmittance and exhibits a large MO effect to obtain high light intensity in MO imaging.

In this study, we developed a new spin-coating process for film formation of the magnetic garnet on a flexible substrate. Figure 2 shows preparing process of magnetic garnet sediment. We prepared the bismuth-substituted yttrium iron garnet (Bi:YIG, Kojundo Chemical Lab. Co., LTD., Kariya, Aichi, Japan). The Bi:YIG particles were prepared by the co-precipitation method [31,32,33,34]. The composition of Bi:YIG was Bi_0.5_Y_2.5_Fe_5.0_O_12_, and the purity of Bi:YIG was 99.9%. A polyvinyl alcohol (PVA, Kishida Chemical Co., LTD, Osaka, Osaka, Japan) aqueous solution was used as the organic binder. PVA concentrations in the PVA aqueous solution were 10 wt% and 15 wt%. Bi:YIG particles were mixed into the PVA aqueous solution with a stirrer at a weight ratio of 1:3 for 24 h. The mixture was deposited on the flexible substrate with the spin-coating method. After spin coating, the substrate was dried at room temperature to form the magnetic garnet sediment. Table 1 shows the film formation condition for a Bi:YIG-PVA sediment.

We investigated the transmittance, magnetic hysteresis loop and crystallizability of a Bi:YIG-PVA sediment on the flexible substrate deposited by this new spin-coating method. Transmittance of the Bi:YIG-PVA sediment obtained was measured with a visible spectrophotometer, the magnetic hysteresis loop was measured with a vibrating sample magnetometer (VSM, Tamakawa Co., LTD, Sendai, Miyagi, Japan) and crystallizability was evaluated with X-ray diffraction (XRD, Rigaku Co., LTD, Akishima, Tokyo, Japan).

## 3. Results

### 3.1. Bismuth-Substituted Yttrium Iron Garnet Film

The bending image of a flexible Bi:YIG-PVA sediment is shown in Figure 3a. The magnetic garnet sediment could be bent freely, and the surface did not crack or break away even after it was bent, remaining in good shape. The thickness of the Bi:YIG-PVA sediment was approximately 30 μm which was estimated from cross-sectional observation by Field Emission Scanning Electron Microscope (SEM, Japan Electron Optics Laboratory Co., LTD, Akishima, Tokyo, Japan) as shown in Figure 3b. We confirmed the crystallization of the Bi:YIG-PVA sediment by XRD. A deposited Bi:YIG sediment showed typical peaks of magnetic iron garnet [35,36] as shown in Figure 3c. Figure 3d shows a hysteresis loop of a flexible Bi:YIG-PVA sediment by VSM. The Bi:YIG-PVA sediment showed a ferromagnetic property.

Figure 4 shows the transmittance spectrum of the magnetic garnet film measured with the visible spectrophotometer. Both Sample 1 and Sample 2 had almost 0% transmittance in the wavelength range from 400 nm to 1400 nm. As shown in Equation (2), MO imaging requires that light pass through it to some extent, and previous studies have shown that a transmittance of about 20% is necessary. There are several possible reasons for the 0% transmittance, such as thickness, multiple scattering and iron absorbed.

### 3.2. Yttrium Aluminum Garnet Film

Because Bi:YIG-PVA sediment had almost 0% transmittance in the visible range, we used yttrium aluminum garnet (YAG) particles, in which the iron sites of Bi:YIG were substituted with aluminum and have a small absorption coefficient, to form the film and investigate its optical properties. The composition of YAG was Y_3_Fe_5_O_12_, and the purity of Bi:YIG was 99.9%. The film formation conditions are shown in Table 2. The transmittance spectrum of the deposited YAG-PVA sediments is shown in Figure 5. Based on the measurement results, we were able to confirm that the YAG-PVA sediment exhibited around 10% to 40% transmittance in all film formation conditions. A distortion in the wave form harmonic around the wavelength of 800 nm was due to noise caused by the detector of the spectrophotometer switching from the visible region to the near-infrared region.

## 4. Discussion

The deposited layered with this process is a discontinuous sediment, wherein the particles are dispersed on the substrate. Therefore, it is expected that the light irradiated to the magnetic garnet particles will undulate by resonance or absorb the particles and behave as shown in Figure 6. In addition, the light intensity measured with the spectrophotometer may include the light passed between particles. Such light contributes to the transmittance but cannot be influenced by the Faraday effect because it does not pass through the magnetic material. Thus, we estimated the ratio of light that passed through the particle by analyzing the density of the magnetic garnet particle on the substrate.

The weight of the particles deposited on the substrate was measured to calculate the density of the magnetic garnet particles. The mass of the samples includes the mass of the substrate and PVA. Therefore, we measured the mass of the substrate before deposition and the mass of the sediment deposited with PVA alone. Then, the mass of the YAG-PVA sediment deposited on the substrate was measured by subtracting the mass of the substrate and PVA from the mass of the YAG-PVA sediment after deposition. The density of the YAG on the substrate was calculated by dividing the mass of the YAG-PVA sediment measured by the surface area of the substrate.

The calculated density of YAG per unit area on the substrate is shown by the solid line in Figure 7. As the rotational velocity of the spin coater increased, the density decreased. This is because the particles depositing on the substrate decreased as the rotational velocity increased to increase the centrifugal force. The density on the substrate became almost constant at a rotational velocity of 3000 rpm or more around 0.35 mg/cm^2^. The calculation result in the case where the particles were densely deposited on the substrate in only one layer is shown by the broken line in Figure 7. The rotational velocity asymptotically approached the broken line with samples of 3000 rpm or more, suggesting that the particles were densely deposited on the substrate at around one layer. Therefore, the void space between the particles on the substrate could be assumed to be around 20%.

Figure 8a shows a bending image of flexible YAG-PVA sediment. The cross-sectional observations of the deposited YAG-PVA sediment by SEM were shown from Figure 8b to Figure 8c. In Figure 8b, it can be seen that the YAG-PVA particles appear to deposit on the substrate in multiple layers, while forming a cluster. Figure 8d shows the results of cross-sectional observation of sample 3 in Table 2. Meanwhile, it was confirmed that the thickness of the sediment decreased as the rotational velocity of the spin coating increased and was almost at the same level at both 3000 rpm and 6000 rpm. These observations closely correspond with the calculated results shown in Figure 7, and the validity of the calculation was confirmed.

Figure 9 shows the Faraday rotation angle of YAG-PVA sediment. The plot of the circle shows only substrate (without YAG-PVA sediment), and the square plot shows the Faraday rotation angle of the YAG-PVA sediment in Figure 9. This Faraday rotation result shows the paramagnetic material property. From the results in Figure 3 and Figure 9, we confirmed that the garnet sediment was formed by a new spin-coating process.

## 5. Conclusions

Magnetic garnet sediment that can be flexibly bent was manufactured with a new spin-coating process. We investigated the transmittance, magnetic hysteresis loop and crystallizability of a Bi:YIG-PVA sediment on the flexible substrate deposited by this new spin-coating method. We deposited YAG-PVA sediment, and transmittance was measured. As a result, a translucency of 40% was observed.

The density of a YAG-PVA particle deposited on the substrate became constant at a rotational velocity of the spin coating of 3000 rpm or more, and it was conceivable that the particle deposited densely in one layer. Based on these results, it became clear that void space between particles on the substrate was around 20%, and that the YAG-PVA sediment showed translucency of 20% or more.

In the future, we are planning to develop a hybrid material, in which some of the iron sites of Bi:YIG are substituted with aluminum, for performing MO imaging.

## Figures and Tables

**Figure 1 materials-15-01241-f001:**
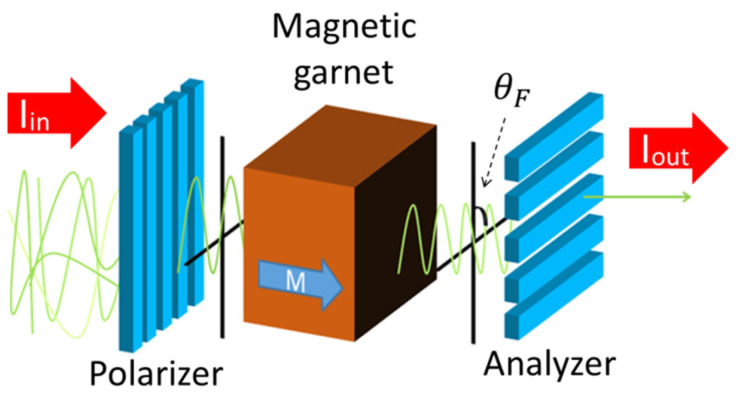
Schematic illustration of magneto-optical (MO) effect.

**Figure 2 materials-15-01241-f002:**
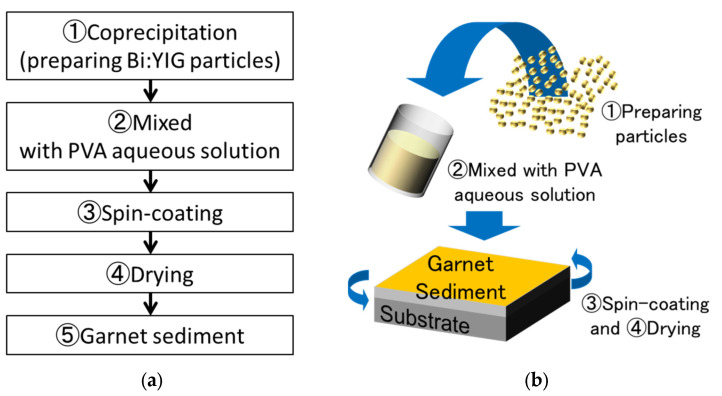
Process of preparing garnet sediment: (**a**) flow of process; (**b**) schematic illustration of spin-coating process.

**Figure 3 materials-15-01241-f003:**
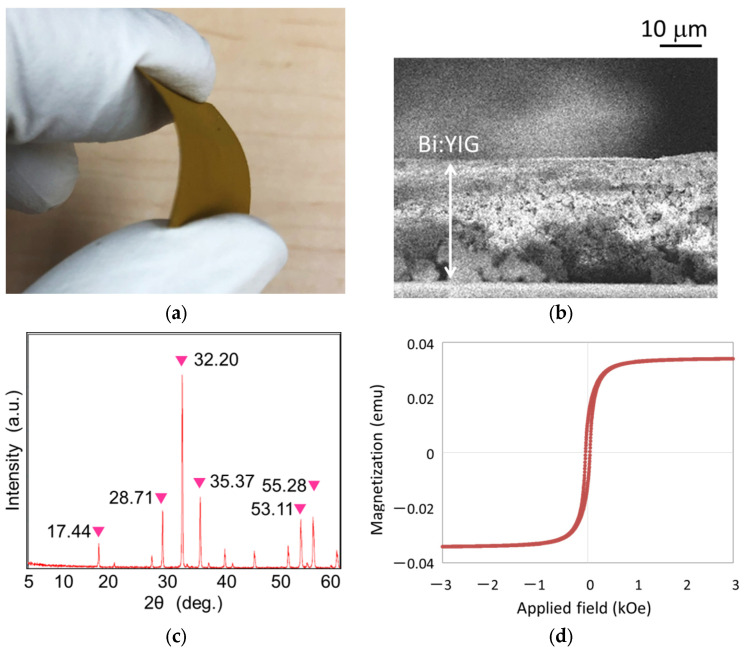
Result of surface examination: (**a**) bending image of a flexible Bi:YIG-PVA sediment; (**b**) cross-sectional observation by SEM; (**c**) X-ray diffraction (XRD) patterns; (**d**) hysteresis loop of a flexible Bi:YIG sediment by a vibrating sample magnetometer (VSM).

**Figure 4 materials-15-01241-f004:**
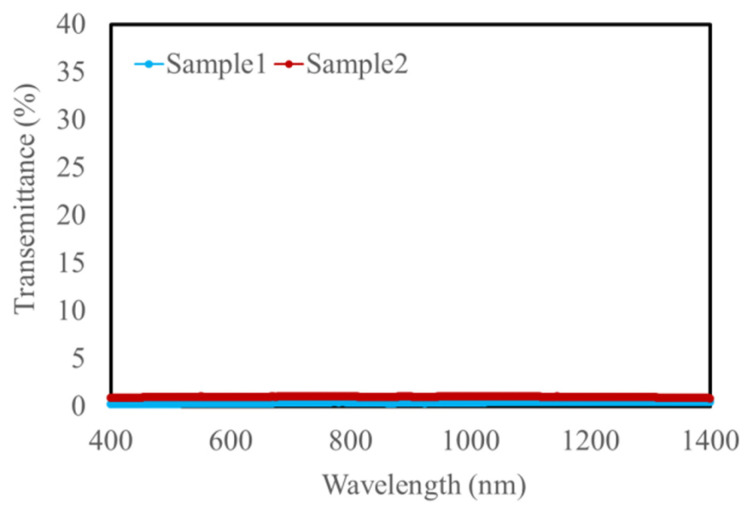
Transmittance spectrum of Bi:YIG-PVA film.

**Figure 5 materials-15-01241-f005:**
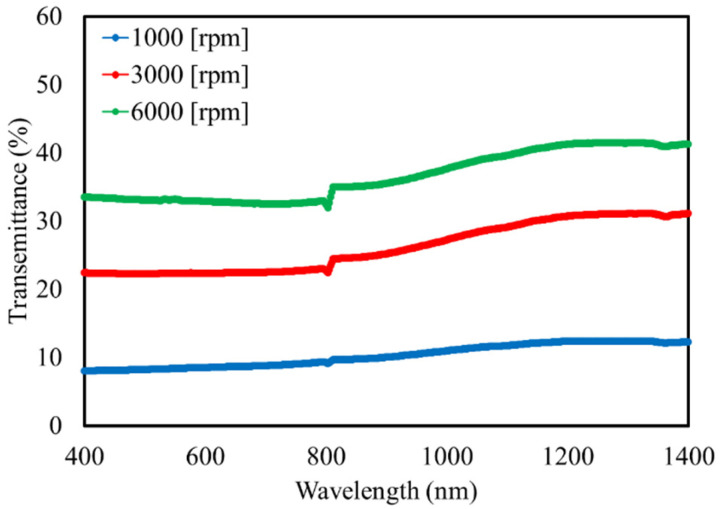
Transmittance spectrum of YAG-PVA film.

**Figure 6 materials-15-01241-f006:**
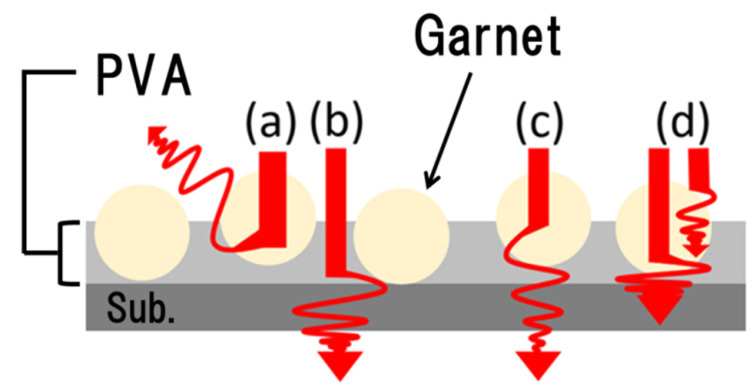
Behavior of the light irradiated to the magnetic garnet film: (**a**) reflected light; (**b**) the light passed between particles; (**c**) transmitted light; (**d**) absorption.

**Figure 7 materials-15-01241-f007:**
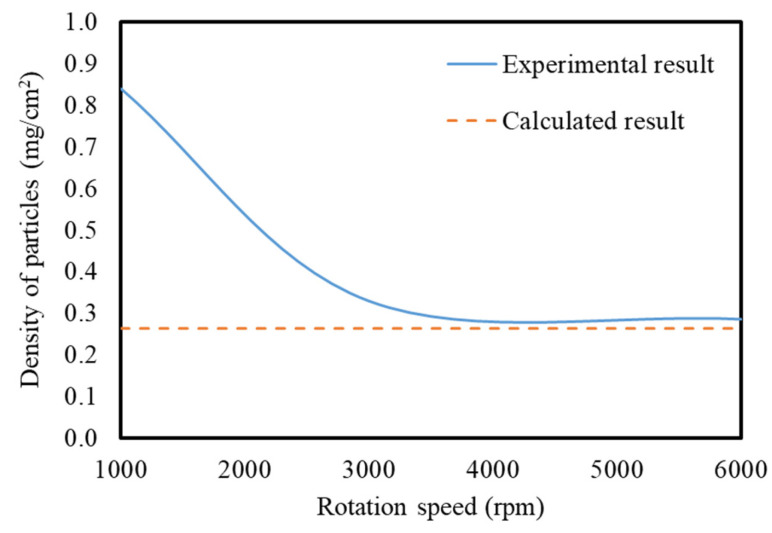
Calculation of the density of the YAG-PVA particle per unit area on the substrate.

**Figure 8 materials-15-01241-f008:**
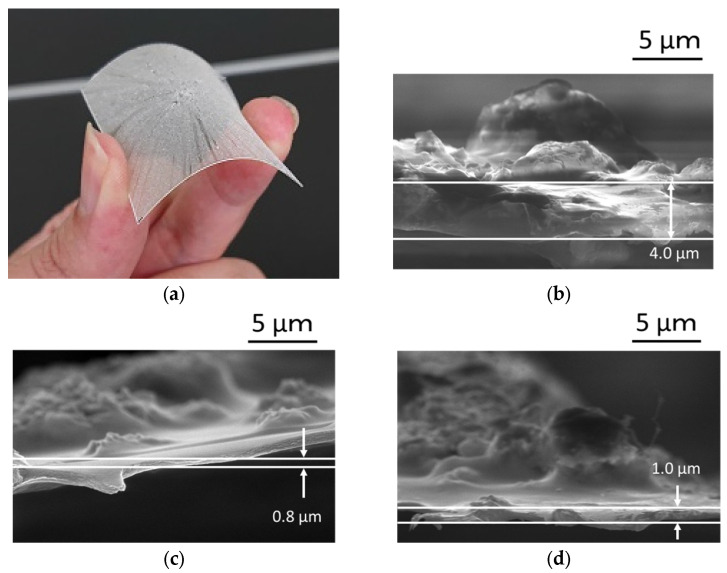
Surface observation of YAG-PVA sediment: (**a**) bending image of flexible YAG-PVA sediment; (**b**) cross-sectional observation of sample 1 by SEM; (**c**) sample 2; (**d**) sample 3.

**Figure 9 materials-15-01241-f009:**
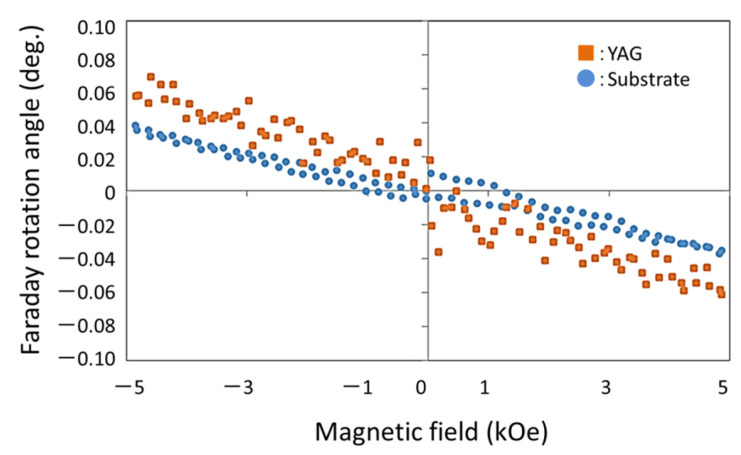
Faraday rotation angle of YAG-PVA sediment.

**Table 1 materials-15-01241-t001:** Film formation condition for bismuth-substituted yttrium iron garnet (Bi:YIG-PVA) sediments.

Sample Number	PVA Aqueous Solution Mass % Concentration (wt%)	Mass Ratio Bi:YIG-PVA	Spin Coating Condition (rpm/sec)	
			Step 1	Step 2
Sample 1	10	1:3	500/10	6000/60
Sample 2	15			

**Table 2 materials-15-01241-t002:** Film formation condition for YAG-PVA sediments.

Sample Number	PVA Aqueous Solution Mass % Concentration (wt%)	Mass Ratio YAG-PVA	Spin Coating Condition (rpm/sec)	
			Step 1	Step 2
Sample 1	15	1:3	500/10	1000/60
Sample 2				3000/60
Sample 3				6000/60

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
