# Peer review of "Properties of Magnetic Garnet Films for Flexible Magneto-Optical Indicators Fabricated by Spin-Coating Method"

_materials, 2022, doi:10.3390/ma15031241_

Round 1
Reviewer 1 Report
The authors present an experimental study on the spin-coating to form magneto-optical films of Bi:YIG-PVA, where bismuth-substituted yttrium iron garnet (Bi:YIG) exhibits high transmittance in a wide visible region and a large Faraday rotational angle while PVA acts as the binder for the magnetic particles in a composite film. The authors report that the resulting film has a flexible mechanical structure (Fig. 3b) and high transmittance (Fig. 5). To describe the transmittance characteristics of the films, the authors provide a model to explain the discrepancy between the experimental dependence of the garnet particle density on the spin coating rotation speed and the calculated results. There are some significant shortcomings that require additional experiments, a major revision of the manuscript and another round of review to justify the publication of this study.
Please see my comments and questions below:
1) I do not see the authors evaluate the previous research on the subject in any detail. A detailed evaluation of the previous studies on spin coating of MO films is needed. In addition, the authors need to justify how this study is different from previous papers on the subject (i.e. flexibility, conformal coating, low thermal budget etc.).
We need to see that the films are magnetooptical, garnet and micro/nanocrystalline:
2) There needs to be a composition and crystallite size analysis (such as from XRD) that gives the content of garnet films.
3) The authors also need to provide magnetic hysteresis loops of the samples.
4) There is no refractive index analysis (i.e. spectroscopic ellipsometry) to evaluate film thickness, n or k.
5) Since the authors do not show any magnetooptical Faraday rotation results, there is no way we can know if the authors report a nonmagnetic or non-magnetooptical film.
6) The authors are also strongly advised to have the manuscript reviewed and edited by a native speaker of English, because there are many grammar mistakes or word omissions starting from the abstract.
These comments need to be properly addressed in order for the study to be sound and acceptable for publication.
Author Response
Dear reviewers and editors:
We appreciate you for taking the time to review our paper and to provide us with useful tips and suggestions for improving it. We have incorporated your suggestions in the best way possible taking into account the comments from all three reviewers. We believe that our new version is more suitable for the journal and should attract more readers. Therefore, we thank you very much for your time, interest, and assistance.
Authors’ action for Reviewer 1.
#1: The previous research on the subject in any detail, - How this study is different from previous papers on the subject.
Yes, the reviewer is right. We checked all sentences of "Introduction", and we rewrite almost all sentences. The originality of this paper is the coating process of the magnetic garnet for flexble MO imaging sensor. We investigated physical properties of garnet sediment of the flexble substrate deposited by this new process. To clarify this and to stress the meaning of this study, we checked all sentences of "Introduction", additionally, added, revised, modified and rearranged some sentences. (from line 38 to line 51)
#2: There needs to be a composition and crystallite size analysis (such as from XRD) that gives the content of garnet films.
We confirmed a crystallization of garnet and we showed XRD patern in Fig.3 (c). And we also showed a composition of Bi:YIG. (line 86)
#3: The authors also need to provide magnetic hysteresis loops of the samples.
We provided a magnetic hysteresis loop of a Bi:YIG sample in Fig.3 (d).
#4: To evaluate film thickness.
We confirmed a film thickness, and we provided a cross-section view by SEM in Fig.3 (b).
#5: magnetooptical Faraday rotation results.
We provided a Faraday rotation of a YAG sample in Fig.10.
#6: English
We checked our English, and we also asked the English proofreading company to check it.

Reviewer 2 Report
The manuscript reports: Properties of magnetic garnet films for flexible magneto-optical indicators fabricated by spin-coating method. Currently, research related to YIG thin films, are based on obtaining excellent quality films on more economical substrates and using economical methods. YIG films with excellent quality are associated with films with low magnetic damping. Usually measurements of ferromagnetic resonance (FMR) are used to verify this property. In the current form, the manuscript is not present sufficient quality, or novel information to be considered for publication in this journal; several changes must be made based on the suggestions below.
1) The summary is weak, very qualitative and does not show the most important novelties of the work.
2) The introduction of relevant background and research progress was not comprehensive enough. Regarding the introduction, the purpose of the work is not clear, and the motivation is absent.
3) The materials and method section is weak; the authors omit valuable information about the synthesis.
4) The purity of raw materials is omitted.
5) Details about film deposition are omitted.
6) There is no clear information in this manuscript that shows the formation of the crystalline phase of YIG with cubic structure. Any information presented here without a demonstration of YIG phase formation may be speculative. Authors should provide information on the structure and phase formation through careful low angle x-ray diffraction analysis.
7) The analysis of figure 3, the result was already expected, even so it was carried out. What do the authors want to show? if this is known? No references were provided by the authors.
8) The authors use the term: magnetic garnet, but there is no analysis to show this important property for YIG films.
9) It may not be appropriate to use the term Film, if you look carefully at Fig 9
10) It is appreciable that the authors have analyzed the Properties of magnetic garnet films for flexible magneto-optical indicators fabricated by spin-coating method, but there is no in-depth scientific discussion. A careful review of the entire manuscript must be performed. Strong scientific discussions must be made throughout the entire manuscript. The authors should improve the quality of the figures. To strengthen the discussions, more up-to-date bibliographic references must be sought and included in the text. More powerful characterizations should be included to strengthen and justify the work.
In general, this paper should be reviewed carefully, attention: careful structural analysis should be done to demonstrate the YIG and YAG phase formation. Without this information, I do not think any results presented are reliable.
Author Response
Dear reviewers and editors:
We appreciate you for taking the time to review our paper and to provide us with useful tips and suggestions for improving it. We have incorporated your suggestions in the best way possible taking into account the comments from all three reviewers. We believe that our new version is more suitable for the journal and should attract more readers. Therefore, we thank you very much for your time, interest, and assistance.
Authors’ action for Reviewer 2.
#1, #2: The summary and The introduction
Yes, the reviewer is right. We checked all sentences of our manuscript. The originality of this paper is the coating process of the magnetic garnet for flexble MO imaging sensor. We investigated physical properties of garnet sediment of the flexible substrate deposited by this new process. To clarify this and to stress the meaning of this study, we checked all sentences of "Introduction", additionally, added, revised, modified and rearranged some sentences. (from line 38 to line 51)
#3: The materials and method section, #5: Details about film deposition.
Yes, the reviewer is right. We added and revised “materials and method” (from line 70 to 98), and we provided a preparing process in Fig.2.
#4: The purity of raw materials is omitted.
The purity of Bi:YIG and YAG was 99.9%. We provided the purity at line 86 and 184.
#6: X-ray diffraction analysis.
We confirmed a crystallization of garnet and we showed XRD patern in Fig.3 (c).
#7: Figure 3, #8: there is no analysis to show important property for samples.
Yes, the reviewer is right. We changed Fig.3. We provided a film thickness, a XRD pattern of Bi:YIG and a hysteresis loop in Fig.3. Additionally, we provided a Faraday rotation angle of YAG sample in Fig.10.
9) It may not be appropriate to use the term Film
We agree this comment. We changed the term from “film” to “sediment”.
10) improve the quality of the figures, the discussions, more up-to-date bibliographic references must be sought and included in the text. More powerful characterizations should be included.
Thank you for your important and constructive comments. We have rewritten our manuscript your suggestions in the best way possible taking into account the comments. We added some updated new figures to provide significant properties of samples. And, we also added new / important reference in the text.

Reviewer 3 Report
Dear Editor,
In the manuscript entitled “Properties of magnetic garnet films for flexible magneto-optical indicators fabricated by spin-coating method” the authors present a fabrication procedure for manufacturing flexible magnetic film from bismuth-substituted yttrium iron garnet (Bi:YIG) and yttrium aluminum garnet. They use a spin coating method, that allows the film to bend without breaking.
I believe that the paper is suitable for your Journal, but I have some questions before my recommendation:
1. I would prefer Fig.3 to be in color, to see if the sample looks black. And a similar photo for the aluminum garnet to see if it is gray or something.
2. What is the thickness of the sample of Fig.4?
3. The authors attribute the zero transmission of Fig.4 to the iron ions. However, it could be because of the thickness. Or, maybe, it could be because of the granular nature of the sample. For example, multiple scattering events between randomly dispersed particles could absorb light (e.g. https://doi.org/10.1088/0957-4484/26/8/085301, https://doi.org/10.1364/OL.36.001884), similarly to what is called “black silicon”.

Author Response
Dear reviewers and editors:
We appreciate you for taking the time to review our paper and to provide us with useful tips and suggestions for improving it. We have incorporated your suggestions in the best way possible taking into account the comments from all three reviewers. We believe that our new version is more suitable for the journal and should attract more readers. Therefore, we thank you very much for your time, interest, and assistance.
Authors’ action for Reviewer 3.
#1: I would prefer Fig.3 to be in color, to see if the sample looks black. And a similar photo for the aluminum garnet to see if it is gray or something.
Thank you for your important and constructive comments. We showed color image in Fig. 3 (a) and also we provided a similar photo of aluminum garnet in Fig. 9 (a).
#2: What is the thickness of the sample of Fig.4?
Thickness is about 30 mm. We provided film thickness from cross-section image by SEM in Fig.3 (b).
#3: The authors attribute the zero transmission of Fig.4 to the iron ions. However, it could be because of the thickness. Or, maybe, it could be because of the granular nature of the sample.
We agree this comment. We added these possibilities in our manuscript. (from line 142 to 143)
Round 2
Reviewer 2 Report
The authors review and improve the manuscript. This new version may be considered for publication.
Reviewer 3 Report
Dear Editor,
I agree with the changes made by the authors, but the language and wording throughout the text must be significantly improved before publication.